# Using a topic model to map and analyze a large curriculum

**Peter A. Takizawa** *

Department of Cell Biology, Yale University School of Medicine, New Haven, Connecticut, United States of America

* peter.takizawa@yale.edu

## Abstract

A qualitative and quantitative understanding of curriculum content is critical for knowing whether it's meeting its learning objectives. Curricula for medical education present challenges due to amount of content, the diversity of topics and the large number of contributing faculty. To create a manageable representation of the content in the pre-clerkship curriculum at Yale School of Medicine, a topic model was generated from all educational documents given to students during the pre-clerkship period. The model was used to quantitatively map content to school-wide competencies. The model measured how much of the curriculum addressed each topic and identified a new content area of interest, gender identity, whose coverage could be tracked over four years. The model also allowed quantitative measurement of integration of content within and between courses in the curriculum. The methods described here should be applicable to curricula in which texts can be extracted from materials.

## Introduction

Medical education at Yale School of Medicine (YSM) comprises three phases: pre-clerkship, clerkship, and advanced training period. In the pre-clerkship period students are taught basic and clinical science and introduced to clinical skills, ethics, and public health. The pre-clerkship period contains over 1000 hours of didactic, active learning and hands-on experiences and relies on the contributions from hundreds of faculty. The amount and diversity of content within the pre-clerkship period challenges leadership to assess how well the content meets the school's educational objectives and to find and evaluate the teaching of specific topics. Further, in an integrated curriculum like that at YSM, the intent is to integrate content from multiple disciplines within a single course based upon the topic being covered. Assessing the success of this integration poses a further challenge for curricular review.

To meet these challenges requires a comprehensive and manageable view of content that allows for qualitative and quantitative analysis. Like most schools, YSM has a curriculum management system to store and organize content. Although these systems are effective for scheduling sessions and distributing materials, they often have limited search capabilities and

**Data Availability Statement:** Corpora for all curricular years analyzed in the manuscript are available here: https://doi.org/10.7910/DVN/CFH8RO.

**Funding:** The author received no specific funding for this work.

**Competing interests:** The author have declared no competing interests exist.

usually cannot generate an overview or summary of the relationships between curricular sessions and content.

In the current analysis, topic modeling was used to understand the organization and relationships of over 700 sessions and their content within the pre-clerkship curriculum at YSM. Session content was mapped to 80 relevant topics using a topic modeling paradigm that uses natural language processing and statistical algorithms to find themes or topics in a large corpus of documents [1]. This paradigm has successfully identified themes in several different domains, including scientific journal abstracts [2], reviews on Yelp [3] and bioinformatics [4].

This paper shows how a topic model can be used to analyze a large curriculum. First, a topic model reduces the complexity of the content from many diverse sessions into a smaller number of topics, which can provide a meaningful overview of a large curriculum. Second, a topic model facilitates mapping content to curricular objectives and quantifies how much of the curriculum addresses each objective. Third, a topic model provides a means to find specific content that may be distributed across a curriculum and to measure how much of the curriculum is covers that content. Lastly, a topic model can measure the integration of content in a curriculum and identify sessions that may be redundant or disconnected.

## Materials and methods

### Processing documents

Documents for all sessions in the pre-clerkship curriculum for a particular class year were collected. The documents were reviewed to remove duplicates and those that did not contain content (e.g seating charts, schedules, etc.). All documents were in PDF format. A program to extract text was written in Python using the package Pdfminer.six [5]. Text was converted to lowercase and non-alphanumeric characters were removed. Texts from documents in the same session were combined. Texts were further processed using the spaCy [6] library to remove stop words, numbers, punctuation. Words were lemmatized to their base forms and only nouns, verbs and adjectives were kept.

### Topic model

Session texts from one class year of the pre-clerkship curriculum were used to generate a topic model. Texts were first processed with the Genism [7] library to identify bigrams (two-word phrases) and trigrams (three-word phrases). Infrequent words (less than five occurrences in the curriculum) and words that were present in more than 40% of the sessions were removed. This final collection of words, bigrams and trigrams was used to generate a corpus for the curriculum.

Topic models were generated using latent Dirichlet allocation (LDA) [8]. Various implementations of LDA exist and to find the one that generated the best model for the curriculum, models were evaluated by measuring topic coherence and by reviewing the most semantically meaningful words in each topic. Topic coherence [9] provides a quantitative measure of how well the words in a topic fit with each other. The Gensim package was used to measure topic coherence in the LDA models. To review the words in topics, the topics for a model were displayed using the LDAvis [10] library which shows the significance of each topic to the curriculum and the 30 most salient words for each topic.

The MALLET [11] package implementation of LDA, which uses Gibbs sampling, generated models with the highest coherence scores and whose topics most often had words seemed more related when reviewed. MALLET LDA is written in JAVA. The Gensim library contains a wrapper written in Python, which was used to generate all topic models. To determine the number of topics that best represent a particular curriculum, topic models were generated for

10 to 150 topics at increments of 10. The quality of each model was evaluated by topic coherence [12] and perplexity on a held out set of the data [13]. For the latter, sessions were randomly divided into five groups. For each number of topics, five LDA models were generated from a combination of texts from four different groups. Each model's perplexity was calculated using the texts from the held out fifth group. Average perplexity for a number of topics was calculated from the perplexities measured for each held out set of texts. Models from each topic number were also evaluated by reviewing the most salient words in each topic.

## Mapping sessions to competencies

A topic was assigned to a one nine competencies based on the 30 most salient words for that topic. The assignment was made by the author who has detailed knowledge of the competencies. Because each session in a topic model receives a score for every topic, a session could then be assigned to one or more competencies based on its topic scores. To quantify how much of the pre-clerkship curriculum is focused on each competency, the scores for each topic in all sessions were summed. This produced a total score for each topic. The scores for topics that addressed the same competency were combined to generate total scores for each competency.

## Measuring content

To determine which sessions had content that was represented by a topic, a document topic matrix was generated which contains topic scores for all sessions in a curriculum. A topic was considered significantly covered in the session if its score for that session was at least 0.1 (maximum score is 1.0). All sessions with scores of at least 0.1 for a topic were linked to that topic in the pyLDAvis model. Double-clicking on the topic shows the sessions whose content is represented by the topic.

To find how content related to gender identity has changed over the past four years, topic models were generated for pre-clerkship curricula from medical school classes of 2021–2024. For each class year, the number of topics for the model was determined by analyzing the distribution of the bigram *gender_identity*. The model with the fewest number of topics in which every instance of the term *gender_identity* partitioned into the same topic was used.

All words from the topic that contained the term *gender_identity* were collected in a spreadsheet. Terms in the topic were reviewed and those that were considered specific for gender identity were kept. Gender identity terms from all four years of were pooled to create a master list of terms. A curricular year's corpus, which contains term counts for that curriculum, was analyzed for number of times each of the gender identity terms appeared in the curriculum.

To find sessions in a curriculum that contained a significant number of gender identity terms, the term counts for each session were analyzed to find the number of times a gender identity term was used. The fraction of gender identity terms in a session was calculated by dividing the total number of gender identity terms in a session by the total number of terms in that session. Sessions where at least 5% of the terms were gender identity terms were considered to have a significant amount of content related to gender identity.

## Integration of content

To identify sessions that presented related content, the similarity between topic scores for all sessions were determined. Two similarity algorithms were considered: cosine similarity [14], which measures the angle between two vectors, and Jensen-Shannon divergence [15], which measures the similarity between two probability distributions. For cosine similarity, two sessions were considered to have related content if their score was 0.6 or more (maximum 1.0). Jensen-Shannon divergence generates from 0 to 1 with identical distributions having a score of

0. For each session, Jensen-Shannon divergence scores were computed for its distribution of topic scores and the distribution of topic score of every other session in the curriculum. The mean and standard distribution of the Jensen-Shannon scores was calculated. To determine a cutoff score below which two sessions would be considered to contain related content, network graphs were generated, as described below, for cutoff scores that were 2, 3, 4, or 5 standard deviations below the mean. A sample of connections between sessions (~20%) were analyzed by the author who has detailed knowledge of the content in those sessions. This review of connections found a cutoff score 4 standard deviations below the mean produced the most accurate set of connections.

A network graph of related session was created using the NetworkX [16] library. To visualize the graph, the pyvis library was used and the title of sessions was added to nodes. Comparing the networks generated by cosine similarity and Jensen-Shannon divergence showed more meaningful connections between sessions using Jensen-Shannon divergence to determine similarity between sessions.

To measure integration of a curriculum, network density and the mean number of connections were calculated. Network density is a built-in function of the NetworkX library and uses the following function:

$$d = \frac{2m}{n(n-1)}$$

In the equation, n is the number of nodes and m is the number of edges or connections in the graph. A graph without connections would have a density of 0, and a graph in which each node is connected to ever other node would have a density of 1. The mean number of connections was determined by summing the connections for every session and dividing by the total number of sessions.

To determine the mean number of connections between sessions within the same course and between sessions in different courses, the connections for each session were separated into those with sessions in the same course and those with sessions in other courses. The mean number of connections for each group was calculated as above.

## Results

### Identifying topics

A topic model generates a small number of topics from the words in a large collection of documents. Topic modeling uses a bag of words format in which the number of times a word appears in a document is counted but the order of the words is ignored. Words are assigned to topics so that the word counts for a document can be approximated by its distribution of topic scores. Several topic modeling algorithms exist, but latent Dirichlet analysis (LDA) has proven effective for identifying topics in a variety of content areas [17–19]. Several implementations of LDA exist, and LDA Mallet most reliably found topics that were semantically meaningful by measures described below.

A topic model was created from the documents for the most recently completed iteration of the pre-clerkship curriculum (Class of 2024, curricular years 2021–2022). The curriculum comprises over 700 sessions divided into 14 courses and taught across 15 months. All documents that were distributed to students in these sessions were collected. Most of the documents were slides or notes. Some documents, such as seating charts or group assignments were eliminated as they had no information on the content presented in the session. All documents had been converted to PDFs prior to distribution to students.

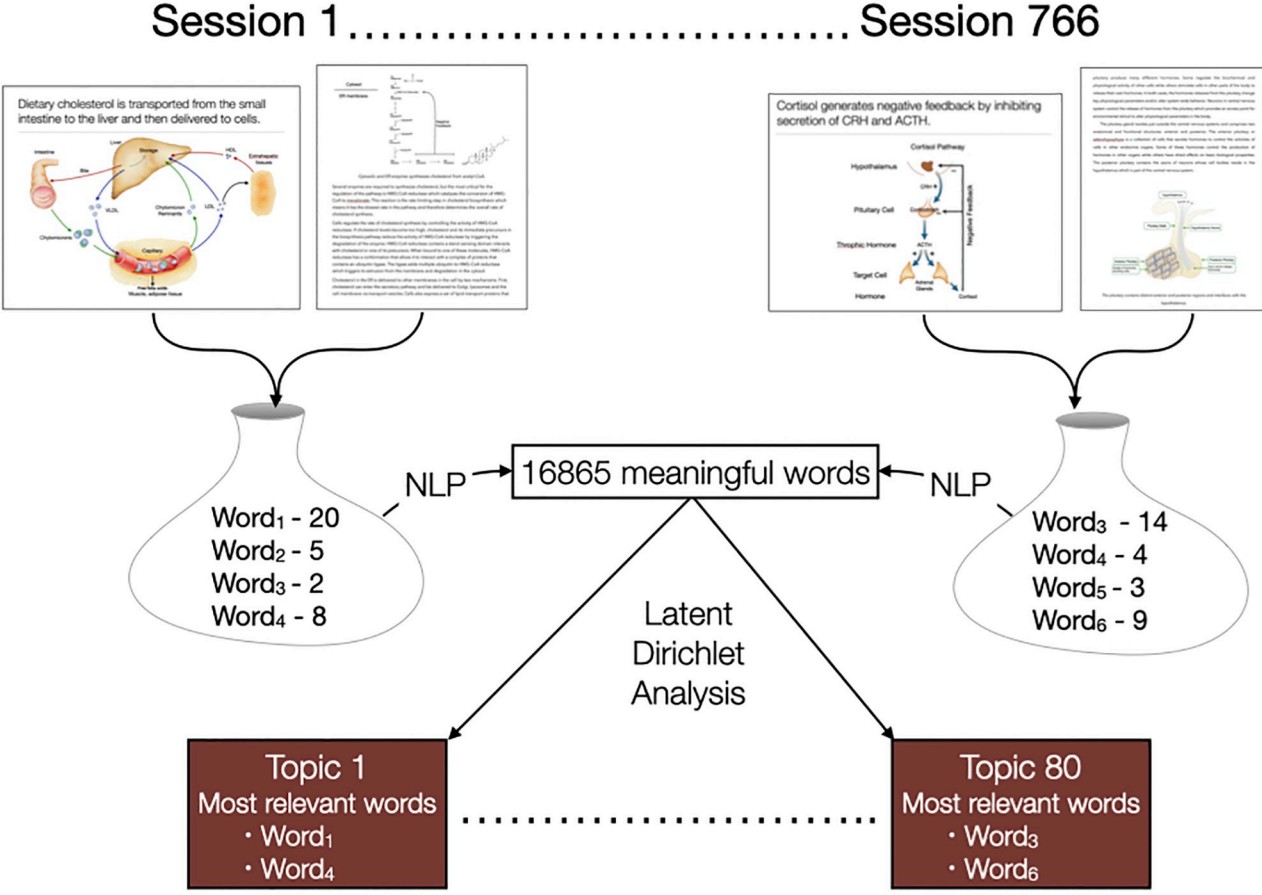

**Fig 1. Extraction and processing of text from curricular documents.** Slides, notes, and other documents were collected for every session in the pre-clerkship curriculum for one medical school class. Text was extracted from the documents using PDF Miner. Text from documents for the same session was combined and then processed to remove non-alphanumerical characters and common words, resulting in a collection of meaningful words. The LDA mallet library was used to generate a topic model.

The first steps in generating a topic model were extracting text from PDFs and processing the text (Fig 1). A program was written in Python using the PDF Miner library to iterate through all documents in the pre-clerkship curriculum, recognize text in the PDFs and store text in a database. The text was processed to remove non-alphanumerical characters and then split into tokens (individual words). Common terms, also called stop words, were removed because they have little semantic meaning. From the collection of tokens, bigrams (two-word phrases) and trigrams (three-word phrases) were identified to create a collection of n-grams called a corpus. The n-grams from documents used in the same session were combined. Thus, topics were created based on sessions in the pre-clerkship curriculum and not individual documents.

The number of topics for a model is a tunable parameter and was determined empirically, using quantitative and qualitative measures. Models were generated and evaluated for a range of topic numbers from 10 to 150 with increments of 10. Topic coherence and perplexity were used to quantitatively measure the quality of the models. Topic coherence measures the semantic relatedness of the words in each topic and provides a quantitative measure of whether the topics in a model make sense. Perplexity measure how well a model predicts the

**Table 1. Most relevant words for select topics.**

| Topic 19 | Topic 36 | Topic 56 | Topic 74 |
|---|---|---|---|
| abortion | action_potential | insulin | interview |
| contraception | channel | glucose | Write |
| hpv_vaccine | membrane_potential | type_diabetes | clinical_reasoning |
| hpv | current | diabetes | Skill |
| abortion_care | depolarization | diabetic | chief_concern |
| induce_abortion | conductance | hyperglycemia | interview_practice |
| cervical_cancer | conduction | inulin_resistance | Step |
| contraceptive | Ion | metformin | social_history |
| iud | membrane | hypoglycemia | clinical_skill |
| pregnancy | sodium_channel | blood_glucose | center_interviewing |

The 10 most relevant words for 4 topics are shown. Relevancy was calculated based on the probability of the word for the topic and the uniqueness of the word to the topic.

words for an unseen set of texts. To measure perplexity, LDA models were generated from 80% of the texts and the held out 20% of texts were used to determine the likelihood that the model predicts the words for the held-out texts. The results show a decline in perplexity score as the number of topics increases (S1 Fig). A lower perplexity score indicates that the model better predicts the words for the held-out texts. Topic coherence varied only slightly for models with topic numbers of 30 or more. A small peak in topic coherence was observed in models with 70 or 80 topics.

Models were also evaluated based on whether the words in the topics suggested a meaningful content area to someone with knowledge of the curriculum. Words are assigned to topics with probabilities, and one approach to identifying the meaning of a topic is to examine the words with the highest probability scores. However, topics are more meaningful to humans when words are displayed based on their relevancy to the topics. Relevancy is calculated from the probability of the word for the topic and the uniqueness of the word to the topic (the fraction of the occurrences of the word in the entire corpus that are in the topic). Empirical studies have identified weights for both factors to generate relevancy scores for words that result in human interpretable topics. Models were considered more representative of the curriculum if the most relevant words in each topic readily led to the identification of the topic. An example of the most relevant words from four topics for a model with 80 topics is shown in Table 1.

Based on its coherence score (0.676), perplexity score and evaluation of relevant words in its topics, the model with 80 topics was selected as more appropriate for the curriculum for the Class of 2024. Although models with more topics had lower perplexity scores, models with lower perplexity have been found to generate topics that are less interpretable by humans [20]. Thus, more weight was given to the topic coherence score. Models with more than 80 topics had similar coherence scores, but some closely related concepts in those models were split into two different topics, which suggested the model had too many topics. In contrast, models with fewer than 80 topics combined two different concepts into the same topic, which indicated that the model had too few topics. The model with 80 topics had the highest coherence score and lowest perplexity score along with the highest proportion of topics whose most relevant words clearly defined a meaningful concept.

To gain a global view of the curriculum, topics were displayed using pyLDAvis library which distributes topics across 2 dimensions based on how close each topic semantically resembles the other topics in the model. The model for the Class of 2024 can be viewed at

http://medcurriculum.org/pre_clerkship_topics. The distribution of topics gives a sense of how much of the curriculum is devoted to different broad content areas. The significance of the topic to the curriculum is proportional to size of the circle. Mousing over a topic circle shows the most relevant words for that topic in the right panel, which allows for a quick identification of the concept represented by the topic.

## Mapping sessions to competencies

YSM has recently adopted a set of competencies that students are expected to meet prior to graduation. A key question is how well the content in the pre-clerkship period matches those competencies. Traditionally, faculty and curriculum leaders would map each session's learning objectives to overall curriculum objectives. For the pre-clerkship curriculum this approach requires human review and mapping thousands of learning objectives from over 700 sessions.

Topic modeling offers a more efficient method to map content in sessions to wider program objectives. A topic model assigns to each session a score for every topic based on how well the words in the topics represent the content in the session. A session's topic scores sum to 1.0. Thus, the content in a session can be quantitatively represented by its distribution of topic scores. To map session content to program objectives, each topic is assigned to a program objective. The topic scores for a session then link the content of that session to program objectives. This approach is more efficient because instead of mapping thousands learning objectives from hundreds of sessions to program objectives, only 80 topics need to be mapped. In addition, the distribution of topic scores for a session provides a quantitatively link between its content and program learning objectives.

To map a topic in the model YSM competencies, the 30 most relevant words for each topic were evaluated to determine the meaning of the topic. Based on its meaning, the topic was assigned to the most appropriate competency (S2 Fig). Next, for each session the scores for topics that mapped to the same competency were summed to generate a score for how well the content in the session addressed the competency. Table 2 shows an example of ten sessions and the fraction of the content in each session that maps to the competencies.

A quantitative measure of how much of pre-clerkship curriculum meets each competency can be obtained from the mapping of individual sessions. For each competency, the scores for that competency from all sessions were summed. The total score indicates the amount of

**Table 2. Mapping of sessions to competencies.**

| Session Title | HP | MTD | CR | PC | PR | CM | RS | PS |
|---|---|---|---|---|---|---|---|---|
| Acute Kidney Injury | 0.00 | **0.89** | 0.08 | 0.00 | 0.00 | 0.00 | 0.00 | 0.02 |
| Alzheimer's Disease | 0.03 | **0.61** | 0.03 | 0.00 | 0.00 | 0.00 | 0.00 | **0.34** |
| Antigen Presentation | 0.00 | **1.00** | 0.00 | 0.00 | 0.00 | 0.00 | 0.00 | 0.00 |
| End of Interview | **0.21** | 0.03 | 0.00 | **0.74** | 0.00 | 0.00 | 0.00 | 0.01 |
| Diarrhea and Intestinal Malabsorption | **0.29** | **0.24** | **0.40** | 0.04 | 0.04 | 0.00 | 0.00 | 0.00 |
| Gender Affirming Surgery | 0.00 | **0.17** | 0.01 | 0.03 | 0.00 | 0.00 | **0.79** | 0.00 |
| Molecular Basis of Beta-Thalassemias | 0.01 | **0.96** | 0.00 | 0.00 | 0.00 | 0.00 | 0.00 | 0.03 |
| Outbreak Investigations and Epidemic Curves | **0.13** | **0.35** | 0.04 | 0.05 | 0.08 | 0.00 | 0.05 | **0.31** |
| Prostate Cancer | 0.05 | **0.52** | **0.12** | 0.14 | 0.00 | 0.00 | 0.01 | **0.15** |
| Race in the Clinical Encounter | 0.03 | 0.06 | 0.01 | 0.02 | **0.15** | 0.10 | **0.48** | **0.15** |

A list of ten sessions in the pre-clerkship curriculum and the fraction of the content that maps to each competency. Scores of 0.1 or higher are bolded. HP: Health Promotion and Disease Prevention, MTD: Mechanism and Treatment of Disease, CR: Clinical Reasoning, PC: Patient Care, PR: Professionalism, CM: Communication, RS: Responsibility to Society, PS: Physician as Scientist.

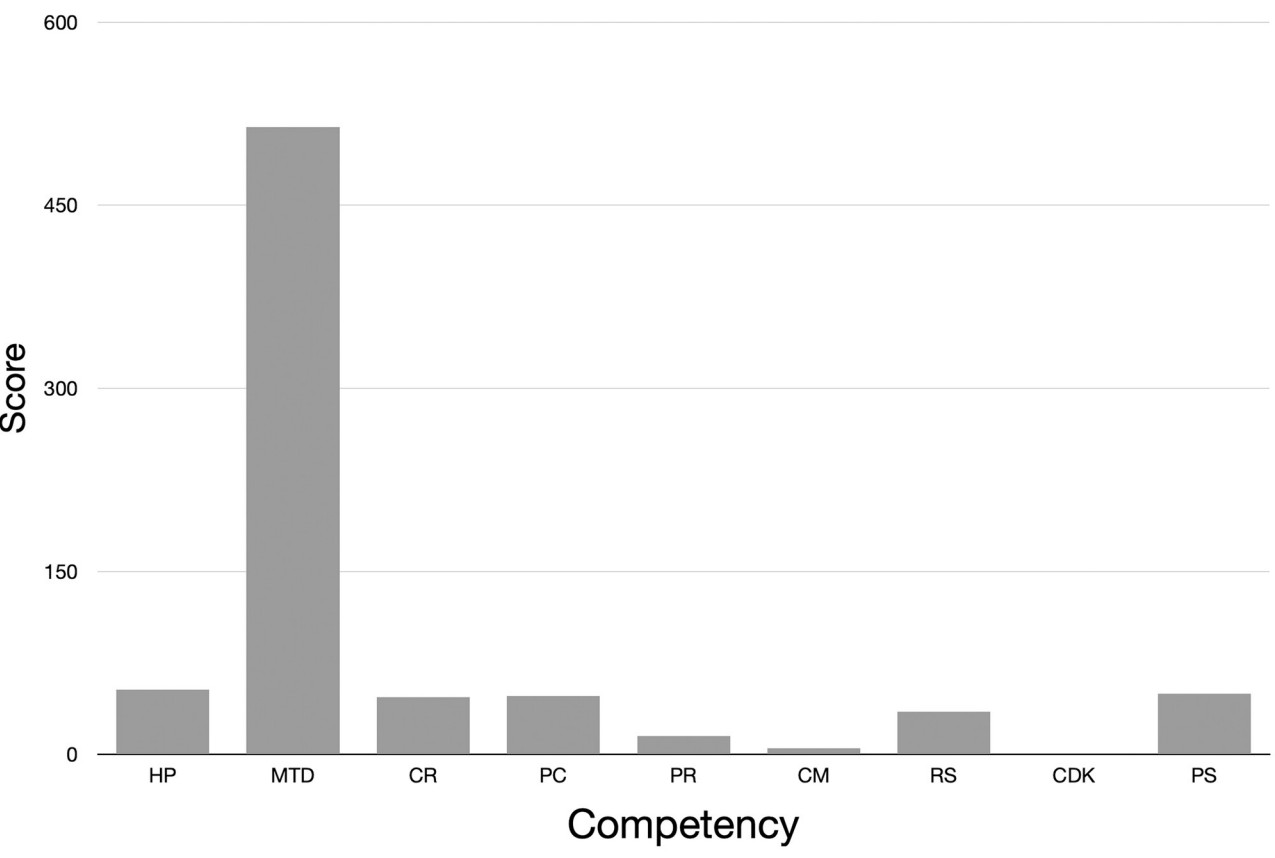

**Fig 2. Quantitative mapping of sessions to competencies.** Each topic was mapped to one competency based on an evaluation of the 30 most salient words for the topic. The scores for how well each topic represented the content for all sessions was summed to generate a total score for each topic. The total scores for topics that mapped to the same sub-competency were summed to generate a final score for each sub-competency. HP: Health Promotion and Disease Prevention, MTD: Mechanism and Treatment of Disease, CR: Clinical Reasoning, PC: Patient Care, PR: Professionalism, CM: Communication, RS: Responsibility to Society, CDK: Creation and Dissemination of Knowledge, PS: Physician as Scientist.

content in the pre-clerkship curriculum that addresses that competency (Fig 2). The results show that most of the content in the pre-clerkship curriculum maps to competencies that address mechanisms and treatment of disease (MTD). This is not unexpected as most of the sessions in the pre-clerkship curriculum present content on basic and clinical science which focuses on the mechanisms and treatment of disease. Some content also maps to competencies around health promotion and disease prevention (HP) and patient care (PC). Curricular leaders can use the mapping to identify curricular objects that are over and underrepresented by the content.

A validation of the model's mapping of content to competencies can be seen by comparing it to the mapping of course objectives to competencies done by course directors (S3 Fig.). The number of course objectives that maps to each competency show a similar distribution as the topic model's mapping. An important difference between the mappings is that each course objective encompasses different amounts of content, and therefore, the number of course objectives that map to each competency may not reflect the total amount of content in the curriculum that maps to each competency. In contrast, the topics were quantitatively generated from the content and may provide a more accurate measure of the amount of content that maps to each competency.

**Table 3. Mapping course content to competencies.**

| Course | HP | MTD | CR | PC | PR | CM | RS | PS |
|---|---|---|---|---|---|---|---|---|
| Across the Lifespan | **0.20** | **0.51** | **0.10** | 0.05 | 0.02 | 0.01 | 0.05 | 0.07 |
| Clinical Skills | **0.21** | 0.07 | 0.05 | **0.53** | 0.02 | 0.00 | 0.09 | 0.02 |
| Homeostasis | 0.02 | **0.84** | 0.06 | 0.03 | 0.01 | 0.00 | 0.00 | 0.04 |
| Professional Responsibility | **0.10** | 0.02 | 0.03 | **0.18** | 0.08 | **0.20** | **0.35** | 0.03 |
| Populations and Methods | **0.10** | 0.05 | 0.01 | 0.01 | 0.04 | 0.01 | **0.15** | **0.64** |

A list of five courses in the pre-clerkship curriculum and the fraction of the content that maps to each competency. Scores of 0.1 or higher are bolded. HP: Health Promotion and Disease Prevention, MTD: Mechanism and Treatment of Disease, CR: Clinical Reasoning, PC: Patient Care, PR: Professionalism, CM: Communication, RS: Responsibility to Society, PS: Physician as Scientist

A similar approach can determine how much content from individual courses and pedagogies maps to program objectives. Sessions were grouped by course, and the competency scores for sessions in the same course were summed to calculate how much of the course's content mapped to each competency. The mapping of five courses to competencies is shown in Table 3.

Likewise, sessions were grouped by pedagogy or teaching method, and the competency scores for the sessions using the same pedagogy were summed to calculate how much content presented by a pedagogy mapped to each competency. The results from six pedagogies are shown in Table 4.

The results demonstrate how a topic model can efficiently map content to program objectives and quantitatively measure the amount of curricular content that addresses each objective. The model also allows an analysis of how different elements of a curriculum, including courses and pedagogies, meet program objectives.

## Measuring how much content addresses a topic

Topic modeling can also quantify how much of the curriculum covers a specific topic or area of content. Because each session receives scores for every topic, sessions whose content addresses a specific topic can be identified by finding sessions that have a score for that topic above a certain threshold. For example, in the topic model for the class of 2024, topic 62 covers content about the heart (most relevant words: valve, mitral valve, aortic valve, heart sound, right ventricle, heart, valvular). Using a threshold score of 0.1, seventeen sessions in the curriculum were found to have content that is represented by the topic. Likewise, topic 59 covers

**Table 4. Mapping pedagogies to competencies.**

| Pedagogy | HP | MTD | CR | PC | PR | CM | RS | PS |
|---|---|---|---|---|---|---|---|---|
| Clinical Correlation | **0.57** | **0.18** | 0.07 | 0.03 | 0.02 | 0.00 | 0.01 | **0.13** |
| Interactive | 0.06 | **0.82** | 0.03 | 0.01 | 0.02 | 0.00 | 0.01 | 0.04 |
| Lab | 0.01 | **0.90** | 0.03 | 0.03 | 0.01 | 0.00 | 0.01 | 0.01 |
| Lecture | 0.07 | **0.70** | 0.03 | 0.05 | 0.02 | 0.01 | 0.04 | 0.07 |
| Team-Based Learning | 0.02 | **0.76** | **0.13** | 0.04 | 0.01 | 0.00 | 0.02 | 0.02 |
| Workshop | 0.07 | **0.49** | **0.13** | 0.13 | 0.02 | 0.00 | 0.08 | 0.09 |

A list of six pedagogies in the pre-clerkship curriculum and the fraction of the content that maps to each competency. Scores of 0.1 or higher are bolded. HP: Health Promotion and Disease Prevention, MTD: Mechanism and Treatment of Disease, CR: Clinical Reasoning, PC: Patient Care, PR: Professionalism, CM: Communication, RS: Responsibility to Society, PS: Physician as Scientist

immunology (most relevant words: antigen, antibody, adapt immunology, lymphocyte, pathogen, immune response, immune system). Based on their score for topic 59, forty-one sessions in the curriculum were found to contain content covered by topic 59. Sessions with a score above 0.1 for a topic are revealed in the topic model for the Class of 2024 by double-clicking on the circle that represents that topic (http://medcurriculum.org/pre_clerkship_topics). This analysis allows curriculum directors to measure how many sessions in a curriculum contain material in a specific content area. In addition, sessions that address a specific topic can be easily found through their scores for that topic.

A topic model can also discover topics or content areas that may not seem significant because they are spread across the curriculum in different courses. For example, the topic model for the Class of 2024 had a topic in which the many of the most relevant words were related to gender identity (gender, transgender, sex, gender identity, identity, sexual, sexual orientation). This topic was interesting because a recent initiative at YSM focused on improving the coverage of gender identity in the curriculum. Course directors were asked to have their faculty consider adding content related to gender identity to existing lectures and small-group sessions, such as workshops and labs. To find sessions that presented some content related to gender identity, sessions were evaluated on whether their score for the gender identity topic was above a certain threshold (0.1). Nineteen sessions spread across nine courses were identified. Traditional search methods may not have found all these sessions because gender identity content was a small fraction of the total content in many of the sessions.

Topic modeling also provided a means to measure whether the amount of gender identity content had increased over four consecutive years of pre-clerkship curricula. Topic models were generated from sessions in the pre-clerkship curricula for class years 2021–2023. For all four topic models (class years 2021–2024), the topic which contained the term *gender identity* was found. To compare the coverage of gender identity across the four consecutive years, terms that seemed semantically related to gender identity were identified. A topic contains hundreds of words, some of which may not be specific for gender identity. For example, the term *person* is found in the gender identity topic but is also found in many other topics. Parsing the terms in the gender identity topic in the four different topic models generated a list of 109 terms that were considered specific to gender identity.

To measure changes in the coverage of gender identity over the four consecutive years of pre-clerkship curricula, the number times a word from the list of gender identity terms appeared in the corpus for each curricular year was counted. The results show an increase in the use of these terms over the past four years (Fig 3). In addition, the fraction of terms in a session that were one of the gender identity terms was calculated for every session in the four curricular years. The number of sessions in which 5% of the terms were related to gender identity has increased over the last four iterations of the pre-clerkship curriculum (Fig 3). The results demonstrate the growth in coverage of content related to gender identity over the past four years.

## Integration

YSM uses an integrated curriculum where courses draw on content from basic science and clinical disciplines to create system and organ-level curricula. During the design of the curriculum, leaders met to identify which content from traditional disciplines that should be covered in each course. Course directors then distributed this content across the sessions in their courses, while trying to retain connections between sessions to create a cohesive course. The curriculum was also designed so that each course builds upon what was covered in previous courses. Thus, some sessions in different courses should cover related content.

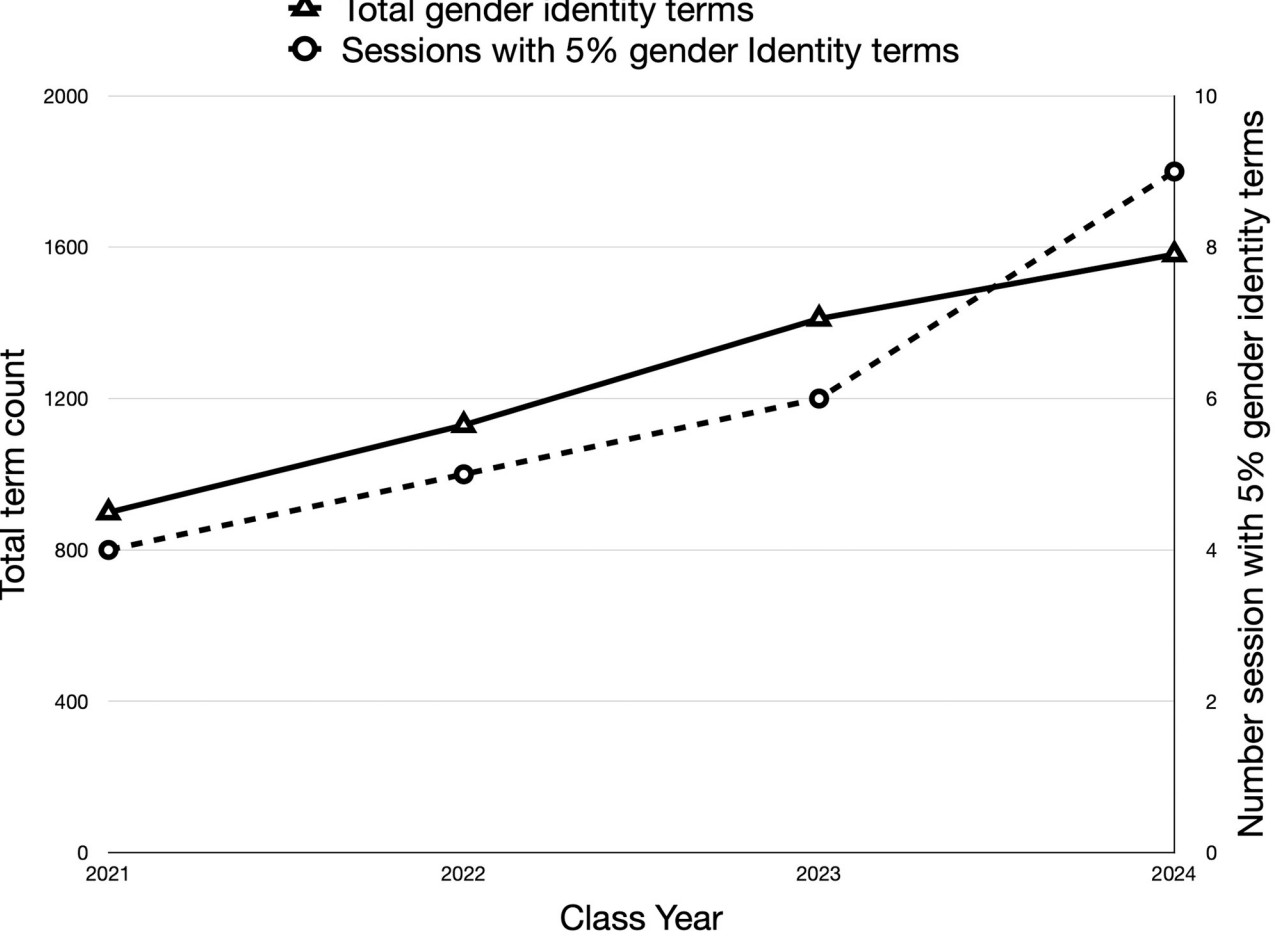

**Fig 3. Change in use of gender and gender identity terms across four years of pre-clerkship curricula.** The total number of gender identity terms used in four years of pre-clerkship curricula (triangles). The number of sessions in which 5% of the terms in the documents were gender identity (circles).

Topic modeling can be used to find sessions that cover similar content within and between courses. As described above, the content in a session can be represented by its probability distribution of topic scores. By comparing the similarity of the topic distributions between two sessions, the relatedness of content in the sessions can be quantitatively measured.

To identify related sessions in the curriculum for the Class of 2024, the similarity of the topic distributions of each session to every other session was calculated using Jensen-Shannon divergence. The algorithm generates a score between 0 and 1 with a lower score indicating greater similarity between two distributions. The mean and standard deviation of the Jensen-Shannon divergence scores were calculated. To determine a cutoff score below which two sessions would be considered to have related content, connections between sessions were generated for cutoff scores 2, 3, 4, or 5 standard deviations below the mean. For each cutoff score, a sample (~20%) of connected sessions were reviewed by the author for accuracy. A cutoff score 4 standard deviations below the mean produced the most accurate connections between sessions.

To visualize the related sessions, a network map of the connections between session was generated (Fig 4A). Each node in the map represents a session, a link or edge between two

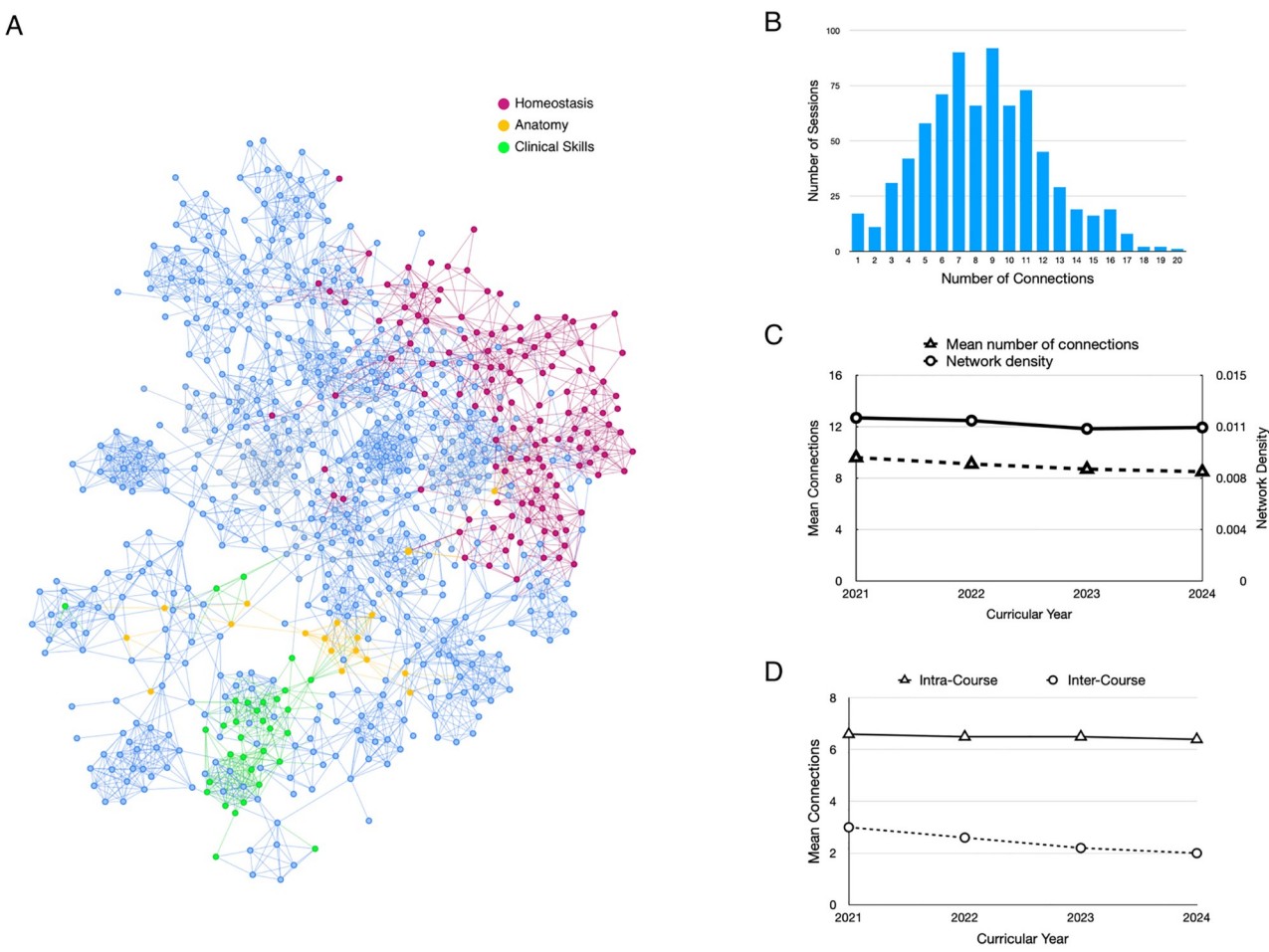

**Fig 4. Integration of content has declined over the past four years.** A. A network map of sessions in the pre-clerkship curriculum. Each node represents a session and a line between nodes indicates the sessions cover related content. Sessions from three courses have been highlighted: Clinical Skills (green), Anatomy (yellow) and Homeostasis (red). The network map can be accessed online at http://medcurriculum.org/pre_clerkship_network. B. Distribution of connections for sessions. The histogram shows the number of sessions that are connected to The map Integration for each class year was determined by the two different metrics of connectivity in network graphs of sessions. Network density (circles) is measure of the fraction of all possible connections in a network graph that exist in the graph. Mean connections (triangles) is the mean number of connections for each session in the graph. Both show a slight decrease over the past four years. B. Integration within courses has remained stable while integration between courses has declined. The mean number of connections between sessions in the same course (triangles) and between sessions in different courses (circles) are shown.

nodes indicates that the sessions cover related content based on the similarity between their topic distributions. The network map was coded so that the nodes in up to four different courses could be labeled to reveal connections between sessions within a course and between courses (http://medcurriculum.org/pre_clerkship_network). The map provides a qualitative measure of the integration of content in the pre-clerkship curriculum.

The network can help curriculum directors identify sessions whose content may need to be reviewed because it is either disconnected from the curriculum or redundant with other sessions. To find those sessions, a histogram of the number of connections per session was generated for the Class of 2024. The results show a binomial distribution with the mean number of connections per sessions at 8.5 and a standard deviation of 3.6. Sessions with many connections (e.g., > 16) may have content that is redundant with the rest of the curriculum and could be eliminated or repurposed to address underrepresented content. The content in sessions

with few connections may need to be reviewed to determine how it can be better integrated with the rest of the curriculum.

The network map can also provide a quantitative measure integration of the entire curriculum. The number of connections between sessions in the map reflects the level of integration of the curriculum. To measure integration of the curriculum, two methods were used to calculate connectivity in the network graph. The first, as described above, was the mean number of connections per session. The second was network density, which is the fraction of all possible connections that exist in the graph. Because the number of possible connections in a network graph is very large, most networks are sparse, meaning that each node is connected to a small fraction of the other nodes in the graph (Fig 4A).

The change in integration over the past four years was also determined. Topic models were generated for the pre-clerkship curriculum in class years 2021–2024. The similarity between sessions in each class year was determined using Jensen-Shannon divergence of topic distributions of sessions in each class year. Network maps were created for each class, and network density and mean number of connections were calculated. The results show a slight decline in network density and mean number of connections over the past four years, which indicates that there are fewer related sessions in the most recent curricular year compared to four years ago (Fig 4C).

To determine whether the decline in related sessions within the pre-clerkship curricula over the past four years is due to a reduction in the number of related sessions within a course or between different courses, the mean number of connections was calculated for sessions in the same course and between sessions in different courses. The results show that connections between sessions within the same course have remained steady over the past four years. Connections between sessions in different courses have declined in each of the last four years (Fig 4D). The results suggest that there are fewer sessions that cover similar content between courses compared to four years ago.

## Discussion

The results illustrate how a topic model can provide qualitative and quantitative understanding of a large and diverse curriculum. By representing the content from a large collection of documents as a smaller number of topics, a topic model provides a manageable overview of the content. A topic model can facilitate mapping content to curricular objectives and quantitate the amount of content that addresses each objective. A topic model can discover topics that are presented as minor components in sessions which are distributed across a curriculum and as a result may not seem significant to curriculum reviewers. Lastly, a topic model can be used to measure the integration of a curriculum and identify sessions whose content is redundant or disconnected with the rest of the curriculum.

Mapping content to broader learning objectives or competencies is critical for evaluating how well a curriculum meets an organization's educational goals and as is the case for medical education, a requirement to meet the standards of licensing agencies. Mapping can be time-consuming and inefficient in a large and diverse curriculum because it requires faculty with specific knowledge of content to perform the mapping. The topic model presented here provided a first approximation of the mapping between individual sessions in the pre-clerkship curriculum and school-wide competencies for Yale School of Medicine. The mapping could be refined by faculty.

The mapping done here is only a first approximation because we are missing data for some sessions in the pre-clerkship curriculum. The data for the model comes from the text in documents for sessions in the pre-clerkship curriculum. Some sessions in the pre-clerkship

curriculum, especially those that focus on clinical skills, don't have documents that describe the content or activities in the sessions. As a result, the content in those sessions is not included in the model. Thus, a small number of sessions are not mapped or may be incorrectly mapped due to lack of information about those sessions. Creating a written description of those sessions would generate a more complete and accurate mapping.

Despite this limitation, the mapping of content to competencies closely paralleled the mapping of course objectives to competencies performed by course directors, which provides some validity of the topic model. Some competencies, such as Communication (CM) and Responsibility to Society (RS), had a much larger ratio of course objectives to topic model score compared to Health Promotion (HP) and Mechanism and Treatment of Disease (MTD). Because course objectives encompass different amounts of content, one explanation for the larger ratio is that the course objectives mapped to CR, and PC cover less content compared to the course objectives mapped to HP and MTD. Resolving these differences will require course directors to generate a more detailed mapping of their course content.

Another source of data that is not captured in the model is what is said during sessions. What is spoken during a lecture is usually similar to the content in the slides and notes, but a faculty member may orally provide details that are not contained in the documents. Yale School of Medicine records most lectures in the pre-clerkship curriculum, and the text of what was said during a lecture could be included in the model by converting the audio in the recordings to text. Discussions in workshops, labs and other small-group sessions are not recorded and cannot be included in the data for the model.

Topic modeling also allowed us to measure changes in content on gender identity over four consecutive years of pre-clerkship curriculum. Locating specific content in a curriculum requires knowing the defining terms. Usually, the most significant terms are easy to identify but minor or new terms can be overlooked. Because topic modeling groups words that are frequently used in the same documents, it can find uncommon terms that are associated with well-known words. For example, a topic model of the pre-clerkship curriculum identified a topic that clearly represented gender identity. The most prominent words in that topic are ones that are easily associated with gender identity, but many of the terms were less common and may have been overlooked through simple keyword searching. By identifying a more complete set of words related to gender and gender identity, the growth of those topics in the curriculum could be more accurately tracked over time.

Topic modeling also allowed us to measure the integration of sessions within and between courses. The integrated curriculum at YSM was designed so that each course presented related content across several disciplines. Topic models of four consecutive years of pre-clerkship curricula showed that the similarity between sessions within a course has remained steady. In contrast, the similarity between sessions across courses has declined over the past four years. One explanation for the decline is that redundant content may have been eliminated. Course Directors meet regularly to discuss content in the pre-clerkship curriculum and find gaps and areas of overlap. The discussions can lead to changes that eliminate content that seems unnecessarily redundant. In addition, a committee periodically reviews all courses and evaluates how the content of each course integrates with the rest of the curriculum. The committee can recommend to course directors that they consider whether redundancy in content servers an educational purpose or should be eliminated.

The results raise the question of the importance of integration between courses. Although integration of content within a course was the major focus of the integrated curriculum, less clear is how content between courses should be tied together and whether a minimum level of integration between courses is important for providing a cohesive and meaningful curriculum. Topic modeling provides a means of finding sessions between courses that cover related

material and gives educators an opportunity to assess whether the similarity between courses is redundant or complementary.

The results illustrate how topic modeling of the YSM pre-clerkship curriculum provided a means of mapping content to competencies, tracking the changes of a specific topic and measuring the level of integration of content. The methods described should be widely applicable to curricula from different disciplines.

## Conclusion

Topic models can provide insights into curricula not easily attained by current means. For example, a topic model can locate content areas within curricula and quantitatively measure the significance of that content area to the curriculum. This information can help schools meet accreditation standards set by licensing boards, which often ask schools to emphasize new content areas in response to changing societal conditions. Curricular leaders usually respond by asking faculty to incorporate new content into their sessions, but finding and tracking where this new content has been added in a large curriculum can be labor intensive and inefficient. Because a topic model considers all words across a curriculum, it can locate related content that is presented in different parts of a curriculum. To facilitate the discovery of a specific topic in a curriculum, a variant of LDA, called keyword assisted LDA, allows a user to add seed words that are germane to a topic of interest [21].

A topic model can also facilitate mapping content to a school's overall learning objectives. School's create learning objectives to meet accreditation standards of licensing boards and the policies and values of the school but demonstrating that the content in the curriculum meets the school's learning objectives is challenging. Most schools map learning objectives from individual components of the curriculum to the overall learning objectives of the school. However, learning objectives at the component level often encompass different amounts of content, which prevents curricular leaders from demonstrating how much of the curriculum meets each of the school's learning objectives. A topic model generates human-interpretable topics from curricular content that can be easily mapped to a school's learning objectives. A topic model also scores each topic based how much of the curricular content it represents. Thus, the mapping of topics to learning objectives generates a quantitative measure of how much of a school's content meets each learning objective. Lastly, a topic model also gives each session scores for every topic, allowing curricular leaders to find the sessions that meet the school's learning objectives and make adjust content as needed.

A critical next step in developing topic models to analyze curricula is to validate the results of the models. An important metric is whether the information produced by a model provides users with an accurate understanding of a curriculum which allows them to make meaningful changes to a curriculum. Although this work does not provide extensive human validation of its topic models, the results found some agreement between the mappings generated by a topic model and course directors. Further testing of the model's validity will come when faculty complete a more detailed mapping of content to competencies. A topic model could also be validated by using it to find a specific content in a curriculum that leadership would like to change, and then using the model to track how the content changes over time. The topic model described here identified content on gender identity and then led to a method to show that the content has increased over the past four years. Future work will test whether a topic model proves useful for locating content of current interest and leads to meaningful changes to the content.

## Supporting information

**S1 Fig. Evaluation of topic models by topic coherence and perplexity.** LDA Mallet was used to generate topic models for texts from the class of 2024. Models were generated for topic numbers from 10 to 150 at increments of 10. The quality of the model at each topic number was evaluated by topic coherence and perplexity on a held-out set of texts. Models with higher topic coherence scores have been found to generate topics that make more sense to human reviewers. Models with lower perplexity scores more accurately predict words in an unseen set of texts.
(TIF)

**S2 Fig. Yale school of medicine competencies.** A list of nine competencies that students are expected to meet before graduating. A major goal is to map content in the curriculum to these competencies.
(TIF)

**S3 Fig. Human mapping of course objectives to competencies has similar distribution as mapping of content done by topic model.** The mapping of content to competency is the same as in Fig 2. Superimposed is the mapping of course objectives to competencies performed by the course directors. Each course director mapped the objectives for their course to the competencies. The results show the mapping combined from all the courses in the pre-clerkship curriculum.
(TIF)

## Acknowledgments

I thank Drs. Fred Gorelick and Michael Schwartz for their invaluable feedback on this work.

## Author Contributions

**Conceptualization:** Peter A. Takizawa.

**Data curation:** Peter A. Takizawa.

**Formal analysis:** Peter A. Takizawa.

**Investigation:** Peter A. Takizawa.

**Methodology:** Peter A. Takizawa.

**Project administration:** Peter A. Takizawa.

**Resources:** Peter A. Takizawa.

**Software:** Peter A. Takizawa.

**Supervision:** Peter A. Takizawa.

**Validation:** Peter A. Takizawa.

**Visualization:** Peter A. Takizawa.

**Writing – original draft:** Peter A. Takizawa.

**Writing – review & editing:** Peter A. Takizawa.

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
