## [Decision Letter · Decision Letter 0]

16 Jan 2023

PONE-D-22-31147Using a topic model to map and analyze a large curriculumPLOS ONE

Dear Dr. Takizawa,

Thank you for submitting your manuscript to PLOS ONE. After careful consideration, we feel that it has merit but does not fully meet PLOS ONE’s publication criteria as it currently stands. Therefore, we invite you to submit a revised version of the manuscript that addresses the points raised during the review process. The reviewers recommend reconsideration of your manuscript following major revision. I invite you to resubmit your manuscript after addressing the comments below.

We look forward to receiving your revised manuscript.

Kind regards,

Vijay Kumar

Academic Editor

PLOS ONE

Journal Requirements:

Reviewers' comments:

Reviewer's Responses to Questions

**Comments to the Author**

1. Is the manuscript technically sound, and do the data support the conclusions?

Reviewer #1: Partly

Reviewer #2: Yes

2. Has the statistical analysis been performed appropriately and rigorously? 

Reviewer #1: Yes

Reviewer #2: Yes

3. Have the authors made all data underlying the findings in their manuscript fully available?

Reviewer #1: No

Reviewer #2: No

4. Is the manuscript presented in an intelligible fashion and written in standard English?

Reviewer #1: Yes

Reviewer #2: Yes

5. Review Comments to the Author

Reviewer #1: Dear Authors,

Congratulations on your paper, "Using a topic model to map and analyze a large curriculum", a thorough analysis of unstructured text data drawn from a medical school curriculum. A novel application of LDA to the problem of curriculum design and review.

Q1. This type of work is very difficult for me to assess as I believe that there is a great deal of value in it, but it does not argue for a specific claim other than that it is possible to apply a certain method to a data set. With respect to this claim you have done an excellent job. The LDA method and subsequent analyses have been applied and explained in depth. However, the validity of the measures (by which I mean, do they mean what you say they mean) has not been investigated. The metrics you report cannot be compared to metrics from anywhere else, for example another corpus of texts. You have not user-tested the metrics you developed so cannot characterize whether they would be useful in course review or construction, or whether there are any unforeseen negative consequences for their use. This does not mean that your research is not valuable, it is a necessary step towards developing new metrics. The question is whether it is publishable on its own, or whether the publication also needs some attempt to validate the approach. I don't believe there is consensus on this, and I have heard very strong opinions on both sides. Unfortunately I don't really know what the answer is though but you should acknowledge the issue and how it could be approached. Network metrics have long been suggested as tools to investigate curricula (see anything by Shane Dawson or Caroline Haythornthwaite) but they never see widespread adoption.

There are also some methodological pieces that could use clarification.

1. The exact way that the topics were mapped to the competencies. Was this a manual process? If so, was it done by a single individual or by several and if by several do you have inter-rater reliability measures?

2. The number of standard deviations from the mean used to determine relatedness of the sessions needs some justification. Four seems like a lot to me, but it may not be, you should say why you chose 4.

Q3. I am not sure to what format would be appropriate to attach you data. The dashboards you have provides through links are likely insufficient unless I am reading them wrong. To be in compliance with the PLOS Data Policy you would likely need to provide access to a minimally processed corpus.

Q4. The manuscript will require substantial copy-editing, there are many typos.

Reviewer #2: It is a well written manuscript with a good attention to details. After reading it with interest, I think this manuscript would need a minor revision to address two points that would improve the robustness of the paper:

1. Please provide cross-validation details for your TMs in the methodology and result/supplementary material. It is crucial that TMs are cross-validated as you used an unsupervised version of TMs.

2. I think a lot of critical inference is being missed without a conclusion section. Please write a conclusion section to present the implication of this study in shaping higher-education policies that can be applied beyond YSM.

3. Some context why MTD has such s significantly high competence score (Figure 2) can be helpful in improving the readability to a broader audience. I think the paper would highly benefit from a table that devise all the curriculum jargons with some contextualised examples.

6. PLOS authors have the option to publish the peer review history of their article (what does this mean?). If published, this will include your full peer review and any attached files.

Reviewer #1: **Yes: **Charles Lang

Reviewer #2: **Yes: **Dr Ramit Debnath

---

## [Author Response · Author response to Decision Letter 0]

1 Mar 2023

Reviewer #1: Dear Authors,  Q1. This type of work is very difficult for me to assess as I believe that there is a great deal of value in it, but it does not argue for a specific claim other than that it is possible to apply a certain method to a data set. With respect to this claim you have done an excellent job. The LDA method and subsequent analyses have been applied and explained in depth. However, the validity of the measures (by which I mean, do they mean what you say they mean) has not been investigated. The metrics you report cannot be compared to metrics from anywhere else, for example another corpus of texts. You have not user-tested the metrics you developed so cannot characterize whether they would be useful in course review or construction, or whether there are any unforeseen negative consequences for their use. This does not mean that your research is not valuable, it is a necessary step towards developing new metrics. The question is whether it is publishable on its own, or whether the publication also needs some attempt to validate the approach. I don't believe there is consensus on this, and I have heard very strong opinions on both sides. Unfortunately I don't really know what the answer is though but you should acknowledge the issue and how it could be approached. Network metrics have long been suggested as tools to investigate curricula (see anything by Shane Dawson or Caroline Haythornthwaite) but they never see widespread adoption.

Response: I recognize the referee’s concern about validating the model. As an initial step toward validating the model, I compared the mapping of content to competency generated by the topic model to mapping of course objectives to competencies done by course directors. This is not a direct comparison because course objectives could encompass different amounts of content, but the results show similar distributions of mapping to competencies between the topic model and what was produced by course directors. The results are shown in Supplemental Figure 3 and described in the Results section. The Conclusion also describes the need to further validate the model. 

  There are also some methodological pieces that could use clarification. 1. The exact way that the topics were mapped to the competencies. Was this a manual process? If so, was it done by a single individual or by several and if by several do you have inter-rater reliability measures?

Response: The topics were manually mapped to the competencies by the author. This information was added to the Methods section.

 2. The number of standard deviations from the mean used to determine relatedness of the sessions needs some justification. Four seems like a lot to me, but it may not be, you should say why you chose 4.

Response: The cutoff score was determined empirically, Network graphs were generated for cutoff scores 2, 3, 4, or 5 standard deviations below the mean. I reviewed about 20% of the connections between sessions for which I have detailed knowledge of the content. From this review, I determined that a cutoff score 4 standard deviations below the mean generated the most accurate network graph. This information has been added to the methods and results sections.  

 Q3. I am not sure to what format would be appropriate to attach you data. The dashboards you have provides through links are likely insufficient unless I am reading them wrong. To be in compliance with the PLOS Data Policy you would likely need to provide access to a minimally processed corpus.

Response: Corpuses for the four curricular years have been made publicly available here:

https://doi.org/10.7910/DVN/CFH8RO

 Q4. The manuscript will require substantial copy-editing, there are many typos.

Response: The revised manuscript has been thoroughly reviewed for typos.

Reviewer #2: It is a well written manuscript with a good attention to details. After reading it with interest, I think this manuscript would need a minor revision to address two points that would improve the robustness of the paper: 1. Please provide cross-validation details for your TMs in the methodology and result/supplementary material. It is crucial that TMs are cross-validated as you used an unsupervised version of TMs.

Response: Cross validation was performed for models with topic numbers from 10 to 150 at increments of 10. The results are shown in Supplemental Figure 1.

 2. I think a lot of critical inference is being missed without a conclusion section. Please write a conclusion section to present the implication of this study in shaping higher-education policies that can be applied beyond YSM.

Response: A Conclusion section has been added to the revised manuscript and the implications of the study on shaping

 3. Some context why MTD has such s significantly high competence score (Figure 2) can be helpful in improving the readability to a broader audience. I think the paper would highly benefit from a table that devise all the curriculum jargons with some contextualised examples.

Response: Because the pre-clerkship curriculum was analyzed, most of the topics were expected to map to MTD (Mechanism and Treatment of Disease). Most sessions in the pre-clerkship curriculum cover basic or clinical science whose content mostly covers the mechanisms and treatments of disease. This explanation was added to the results sections. I have tried to identify curriculum jargon in the text and more clearly define the jargon in the text, but if there are terms that still seem unclear, I can define those, too. Supplemental Figure 2 lists all the competencies with a brief description of each.

---

## [Decision Letter · Decision Letter 1]

3 Apr 2023

Using a topic model to map and analyze a large curriculum

PONE-D-22-31147R1

Dear Dr. Takizawa,

We’re pleased to inform you that your manuscript has been judged scientifically suitable for publication and will be formally accepted for publication once it meets all outstanding technical requirements. You will find the reviewers' comments at the end of this email.

Kind regards,

Vijay Kumar

Academic Editor

PLOS ONE

Additional Editor Comments (optional):

Reviewers' comments:

Reviewer's Responses to Questions

**Comments to the Author**

1. If the authors have adequately addressed your comments raised in a previous round of review and you feel that this manuscript is now acceptable for publication, you may indicate that here to bypass the “Comments to the Author” section, enter your conflict of interest statement in the “Confidential to Editor” section, and submit your "Accept" recommendation.

Reviewer #1: All comments have been addressed

Reviewer #2: All comments have been addressed

2. Is the manuscript technically sound, and do the data support the conclusions?

Reviewer #1: Yes

Reviewer #2: Yes

3. Has the statistical analysis been performed appropriately and rigorously? 

Reviewer #1: Yes

Reviewer #2: Yes

4. Have the authors made all data underlying the findings in their manuscript fully available?

Reviewer #1: Yes

Reviewer #2: No

5. Is the manuscript presented in an intelligible fashion and written in standard English?

Reviewer #1: Yes

Reviewer #2: Yes

6. Review Comments to the Author

Reviewer #1: Comments were addressed as requested. Methods sections was updated: topics were manually mapped to the competencies by the author, cutoff score explanation was included, corpus has been made available online on Harvard Dataverse, cross validation metrics were included. Copyediting was completed.

Reviewer #2: Good efforts in addressing the comments. I think the manuscript reads good now and is ready for the next stage.

7. PLOS authors have the option to publish the peer review history of their article (what does this mean?). If published, this will include your full peer review and any attached files.

Reviewer #1: No

Reviewer #2: **Yes: **Dr Ramit Debnath

---

## [Editor Report · Acceptance letter]

11 Apr 2023

PONE-D-22-31147R1 

Using a Topic Model to Map and Analyze a Large Curriculum 

Dear Dr. Takizawa:

I'm pleased to inform you that your manuscript has been deemed suitable for publication in PLOS ONE. Congratulations! Your manuscript is now with our production department. 

Kind regards, 

on behalf of

Dr. Vijay Kumar 

Academic Editor

PLOS ONE